# Effects of Mixing Ratio of Hybrid Carbonaceous Fillers on Thermal Conductivity and Mechanical Properties of Polypropylene Matrix Composites

**DOI:** 10.3390/polym14101935

**Published:** 2022-05-10

**Authors:** Kwan-Woo Kim, Woong Han, Byung-Joo Kim

**Affiliations:** 1R&D Office 1st, Korea Carbon Industry Promotion Agency, Jeonju 54852, Korea; 01090063344@kcarbon.or.kr (K.-W.K.); shareyi@kcarbon.or.kr (W.H.); 2Department of Carbon-Nanomaterials Engineering, Jeonju University, Jeonju 55069, Korea

**Keywords:** carbon fiber, thermal conductivity, composite, carbon filler, polymer, Charpy impact test

## Abstract

This study investigated the effects of carbon fibers and graphite flakes on the composite materials’ heat dissipation properties and mechanical strength with various hybrid ratios in the matrix. Carbon fibers and graphite flakes with high thermal conductivity showed efficiency in heat dissipation performance, and mechanical strength was reinforced by carbon fiber. However, the heat dissipation performance and mechanical strength were greatly changed according to the mixing ratio. The optimal filler mixing ratio was derived for inducing the enhanced physical properties of the composites reinforced by hybrid fillers with different shapes.

## 1. Introduction

The rapid development of the electronic product packaging industry has increased the demand for high-performance composites superior to conventional materials, with lightweight, high mechanical strength, high thermal conductivity, and high electromagnetic interference shielding properties [1,2,3,4]. In particular, the market’s demand for high mechanical strength, heat dissipation properties, weight reduction, and easy and cost-effective ways to produce parts has increased the need for thermoplastic composites using carbonaceous fillers [5,6].

Carbonaceous fillers generally contain carbon fibers, carbon nanotubes, carbon black and graphite powder, and each filler has various shape characteristics such as linear, spherical, and plate-shaped [7]. Therefore, it is possible to form an optimal heat transfer network and stress propagation path by varying the composition ratio of each filler in the matrix [8,9,10,11].

Liu et al. [12] reported the superiority of a heterogeneous filler reinforcement system by manufacturing a microcapsule reinforced with plate-shaped graphene oxide and linear carbon nanotubes to improve the thermal conductivity of the phase-change materials. Sohrab et al. [13] also used plate-shaped graphene sheets and spherical carbon blacks to increase electrical and thermal conductivity simultaneously and confirmed that the thermal and electrical properties of the composite materials were significantly improved at a specific ratio.

Studies have also been conducted on improving thermal and mechanical properties using hybrid carbon nano-fillers in various applications [14,15,16,17,18], but the use of carbon fibers and graphite powder has not been sufficiently researched from the perspective of various commercial and producibility aspects.

Polymers containing carbon fibers, which are conductive fibers, have the advantage of enhancing mechanical properties while improving electrical and thermal conductivity, but require a large amount of carbon fibers to obtain high electrical and thermal conductivity [19,20,21,22]. Therefore, good electrical and thermal properties can be obtained, but when extrusion or injection molding is used, it may be difficult to produce a composite material having a high volume fraction. In addition, using a lot of carbon fiber means that the production cost is high due to the high price of carbon fiber, and this problem can be solved by adding graphite with low price and high electrical and thermal conductivity.

The Hybrid Filler system using linear carbon fiber and plate graphite powder can effectively form a reinforcing network inside the composite material even with a small amount of filler [23,24], and simultaneously manage the cost and mechanical and thermal properties of the composite material. Therefore, it can be a suitable manufacturing method for the essential components of the market [25].

In this study, a thermoplastic (polypropylene) heat dissipation composite was prepared by adding carbon materials (carbon fibers, graphite flakes) with high thermal conductivity in various ratios. The mixing ratio of the hybrid filler with improved thermal conductivity and physical properties of the manufactured composite was investigated.

## 2. Experimental Section

### 2.1. Materials

Polypropylene (PP, SJ-150) was purchased from Lotte Chemical Corporation (South Korea), and the density and melting temperature of the PP were 0.9 cm^3^/g and 160 °C, respectively (from the technical sheet of the supplier). Carbon fibers (CF, rod-shape filler) and graphite flakes (plate-shape filler) were chosen as different fillers for the hybrid filling system, to form the best thermal network and filler distribution in the matrix. The CF used in this work was supplied by Hyosung Advanced Materials (12K, H2550, Jeonju, Korea). The tensile strength and modulus of the fibers were 5.5 GPa and 250 GPa, respectively (the supplier provided these values). The average diameter and density of the fibers were also measured to be 7 μm and 1.8 g/cm^3^, respectively. To make a composite with a thermoplastic resin, carbon fibers were cut with a unit size of 1 inch. Graphite flakes (GF, purity 99.99%) were provided from Aldrich Co., USA, and the density and particle size were 1.9 cm^3^/g and 100 mesh (>150 μm), respectively. Some of the physical and thermophysical properties of the fillers and matrix are listed in Table 1.

### 2.2. Sample Preparation

To prepare the composite, PP, CF, and GF were mixed with a tailored mixing ratio and melt-blended at 160 °C at a mixing chamber temperature with a screw speed of 70 rpm for 30 min in a lab-scale homemade internal mixer. The total weight ratio of matrix to fillers was fixed at 2:1. Table 2 presents the various mixing ratios of the PP/CF/GF in the composites. After melt-blending, each sample was molded by the hot press with a vacuum bag molding method. Processing temperature, time, and pressure were controlled to 170 °C, 15 min, and 10 MPa, respectively.

#### 2.2.1. The Density of the Composites

The density of the composite samples was calculated using two methods, and each value was compared, as illustrated in Figure 1. The first was the volume density based on the Archimedes method [26,27], and the second was the theoretical density using the mixing ratio of the matrix and fillers. According to the Archimedes method, the volume density of the samples was calculated using Equation (1):(1)$$\frac{W_{d}\cdot D_{l}}{W_{S}-W_{w}}$$
where *W_d_* is the weight of the sample dried for 12 h or more, *W_s_* is the sample weight when the liquid probe fills the open pores of the samples completely, and *W_w_* is the weight of the sample in the liquid probe. *D_l_* is the density of the liquid probe.

In order to accurately obtain each value, the composite sample was dried for 12 h at 80 °C under 0.01 kPa, and then weighed; this was referred to as W_d_. The composite sample was positioned in boiling water (liquid probe) for 2 h to remove the air present in the open pores of the sample. After sufficiently replacing this space (pores) with distilled water, it was cooled to 4 °C to obtain an exact *W_w_*. *W_s_* was also carefully measured after removing the moisture remaining on the surface of the sample. The density of the distilled water at 4 °C was 0.99998 g/mL.

Second, the theoretical density was determined based on the mixing amount of fillers and the matrix in the composites [28]. The reported densities of PP, CF, and GF at room temperature (25 °C) were 0.9, 1.8, and 1.9, respectively, as listed in Table 1. The density of each composite sample was calculated according to the mixing ratio in Table 2 by Equation (2):(2)$$\frac{F_{a}\cdot D_{a}+F_{b}\cdot D_{a}+F_{c}\cdot D_{a}}{F_{a}+F_{b}+F_{c}}$$
where *F_a_*, *F_b_*, and *F_c_* are the weight ratios of each element constituting the composite samples, and *D_a_*, *D_b_*, and *D_c_* are the theoretical density values of each element.

#### 2.2.2. Thermal Conductivity, Specific Heat, and Thermal Diffusivity of the Composites

The thermal conductivity, thermal diffusivity, and specific heat of the composite samples were measured via the transient plane source (TPS) method using the Hot Disk instrument (TPS 2500S, Hot Disk Inc., Göteborg, Sweden), and the results are exhibited in Figures 2 and 3. The conductivity values in the perpendicular and horizontal directions were measured, respectively.

A nickel coil wrapped in polyimide film (Kapton) with a diameter of 6.4 mm (#5501) was used as the measuring probe for the thermal conductivity and thermal diffusivity of the composite sample, reducing the error caused by the anisotropic fillers. The estimated data reproducibility and accuracy of the equipment provided by the manufacturer were better than 1% and better than 5%, respectively. The specific heat value of the composite samples was also measured using a highly conductive golden cell as the reference, and the results were compared with the values of the supplier tech sheet. In all experiments, the measuring instrument was pre-heated for 30 min or more to ensure analysis reliability, and the measurement environment was maintained at 25 °C and 30% of relative humidity.

All specimens were machined as flat as possible using an automatic polishing device. Polishing was carried out in steps of 5 min using sandpapers of 1000, 2000, and 4000 grades. All samples were placed about 6 cm from the center of the automatic polishing machine, and the conditions of 50 N and 150 rpm were equally applied. The finished specimen was uniformly cut to 30 × 30 × 3 mm.

Ansys analysis (Ansys Inc., Washington, DC, USA) was done with these three designs to find the better one in terms of heat dissipation. The ambient temperature is taken to be 25 °C, and it is applied to the outer boundary of the air domain. The solver deck is used at default settings with the program executed in one step and maximum iterations to be 1000 with solver tolerance to be 10^−4^. The models have meshed with the tetra type of elements, and the simulation plots are shown in Figure 4. The other various parameters were also selected to get the most accurate result. Many iterations were done to get the proper size of the element to obtain the minor differences between consecutive results and be more accurate.

#### 2.2.3. Charpy Impact Strength of the Composites

An impact strength test was conducted to observe changes in the mechanical strength of the composite with various filler combinations. A Charpy (CEAST^®^ Resil Impactor, CEAST, Norwood, OH, USA) pendulum impact test was employed, according to ASTM D6110 [29], to observe the total energy value required until the final fracture of the composite material. The results are exhibited in Figure 4.

#### 2.2.4. Morphology Analysis of the Cross-Section

The cross-section morphology of the composite samples was characterized using scanning electron microscopy (SEM, AJS200C, 15 kV, Seron Tech. Inc., Uiwang-si, Korea) and shown in Figure 5. The samples were coated with gold (3 mA, 180 s) to enhance the image resolution and prevent electrostatic charging. The surface of the cross-section was washed with ethyl alcohol diluted to 10%, and impurities such as dust were sufficiently removed and dried before analysis.

## 3. Results and Discussion

### 3.1. The Density of the Composites

Figure 1 exhibits the density value of the composite according to the mixing ratio of the hybrid filler. The theoretical density values, based on the data provided by the suppliers and the measured values using the Archimedes method, were compared. The densities of the carbon fiber and graphite powders were 1.8 and 1.9 g/cm^3^ (the data from the suppliers), respectively, with very similar values. Since the total filler content in the matrix was kept the same at 50%, it was confirmed that the composite material had very similar density values, both theoretically and practically. The average density value of the composites was found to be 1.05~1.07 g/cm^3^, and the measured values were very slightly lower than the theoretical values in all samples. Factors, such as internal pores that may be included in the process of manufacturing the composite materials, were not considered in the theoretical value.

In the sample with PP:CF:GF of 100:10:40, the theoretical and measured values were observed to be very similar, suggesting that the generation of pores formed inside the composite material was related to the amount of graphite powder. On the other hand, the difference in density was found to be the largest in the sample with PP:CF:GF of 100:20:30. This is because if the dispersion of the two fillers is not sufficient (meaning that they may form agglomerations due to the dimensional phase separation) in the composite material, the molten matrix resin insufficiently penetrates between the fillers, resulting in the formation of internal pores.

### 3.2. Thermal Conductivity of the Composites

Figure 2 shows the thermal conductivity values in the perpendicular and horizontal directions of the composite, according to the mixing ratio of the carbon fiber and graphite powder. The thermal conductivity values in the horizontal direction were observed to be higher than in the vertical direction in all samples, which was considered to be due to the fact that a large number of fillers were oriented horizontally by the flow of the molten resin during hot compression molding while making the composite materials.

On the other hand, it was observed that the thermal conductivity of the composites was the highest in the 30:20 sample and decreased again. This same trend was confirmed in the vertical and horizontal directions. Considering that the thermal conductivity of the polypropylene is very low (around 0.1 to 0.2 W/mK), the effect of the filler is responsible for enhancing thermal conductivity (3~14 W/mK in the horizontal direction).

Since the density of the two fillers is very similar, it is not necessary to consider the volume fraction in the composite material; therefore, it can be determined that this is due to the geometric characteristics. If the effect of carbon fiber, which is a long and fibrous shape that is advantageous for heat conduction, was dominant, the thermal conductivity should increase as the content of carbon fiber increases. However, it can be confirmed that the carbon fiber content is not a single contribution by seeing that the carbon fiber content decreases after 30 phr in the matrix.

A previous report [30] confirmed that when carbon nanotubes and graphite platelets were mixed, electromagnetic interference shielding properties significantly improved at a specific ratio of the two fillers. That is, when mixing different types of fillers with different shapes, the dispersion degree of the fillers can be significantly enhanced at a golden mixing ratio so that a pathway for electric conduction is effectively formed.

In this study, the thermal conductivity of the 30:20 sample increased more than three times that of the 40:10 sample, which means that this ratio is the golden ratio, at which the dispersion of the filler is optimized. When the number of GF increased beyond 30:20, thermal conductivity decreased. This is believed to be because the optimal ratio of the linear and planar filler ratios was broken, and the internal heat-transfer path was lost.

### 3.3. Thermal Diffusivity and Specific Heat of the Composites

In order to observe the correlation between the composition ratio of the hybrid filler in the composite materials and the thermal properties of the composites in detail, the thermal diffusivity, and specific heat values of the composites were measured and are exhibited in Figure 3.

Thermal diffusivity is an essential factor in determining thermal conductivity and refers to the speed of heat movement inside the medium. In composite materials, it is greatly influenced by the uniformity of the material, that is, the degree of dispersion of the filler. In Figure 3, it was found that the thermal diffusivity was dramatically enhanced in the 30:20 sample and showed a decrease in the 20:30 and 10:40 samples. This is considered to be because the most homogeneous composite material was produced in the 30:20 sample due to the uniform dispersion of the hybrid filler.

All materials have their specific heat value, and in the case of composite material, the specific heat value is determined by the composition ratio of the matrix and the filler, by the principle of the role of the mixture when no closed pores are present inside. However, the specific heat value can be significantly increased when internal closed pores are formed by the non-homogeneous dispersion of the fillers in the composite material.

Figure 3 exhibits the practically measured specific heat value and the calculated specific heat value of the composite material according to the mixture, respectively. The calculated specific heat value of the composite material showed an average of 1.45 J/gK, while the measured specific heat value was observed to be very different depending on the mixing ratio of the hybrid filler. The calculated and measured values were very similar for the 30:20 and 20:30 samples, while the measured values for the 10:40 and 40:10 samples were observed to have very high values compared to the calculated ones. This is recognized to be due to the formation of closed pores inside the composite material, and it is assumed that the hybrid filler was not uniformly dispersed in the 10:40 and 40:10 samples.

As shown in Figure 1, there was not a big difference between the theoretical and the calculated values of the density of each composite material, while the specific heat value in Figure 3 showed a big difference. It is estimated that the degree of dispersion of the filler and the degree of closed pores simultaneously influenced the specific heat value of the composites.

Figure 4 shows the thermal analysis simulation of product design through heat-transfer analysis in the Ansys workbench. As can be seen from the thermal analysis simulation, aluminum products and polypropylene composites containing hybrid carbon fillers show almost the same high heat dissipation performance. It was confirmed that high thermal conductivity was applied as the hybrid carbon filler was added to the polymer, and it was confirmed that it could be developed as a product with weight reduction compared to aluminum products.

### 3.4. Mechanical Strength of the Composites

Figure 5 shows the Charpy impact strength values of composites manufactured by varying the mixing ratio. It was confirmed that the shape of the reinforced filler dramatically influences the impact strength of the composite material. The highest value was the 40:10 sample, containing 40 phr of carbon fiber, which had a value of about 4.3 kJ/m^2^. This gradually decreased as the ratio of carbon fiber decreased. In the case of the 30:20 sample, which had a relatively uniform filler dispersion, when the carbon fiber content decreased 10 phr compared to the 40:10 sample, the impact strength value decreased by 0.19 kJ/m^2^.

Meanwhile, in the 20:30 sample, the content of carbon fiber was equally reduced by 10 phr compared to the 30:20 sample. A significant decrease in impact strength of 0.54 kJ/m^2^ was observed due to the complex effects of low carbon fiber content and poor dispersion of fillers.

### 3.5. Cross-Section Morphology of the Composites

After the Charpy impact test, the fracture section of each sample was observed by scanning electron microscopy, and the results are shown in Figure 6. In the case of the 40:10 sample, a large amount of carbon fiber was observed, and the graphite powder was only partially found. Meanwhile, the aggregation of graphite power was observed in the 20:30 sample as the content of the graphite filler increased. This observation result is considered to be in line with the uniform dispersion result predicted in Figure 2 and Figure 3.

## 4. Conclusions

In this study, the effects of various reinforcement ratios of one-dimensional carbon fiber and two-dimensional graphite powder on the thermal and mechanical properties of polypropylene matrix composites were investigated. When the ratio of carbon fiber and graphite powder was 30 phr and 20 phr, respectively, the highest thermal conductivity and thermal diffusivity were observed, and the lowest specific heat value was observed. It was also confirmed that thermal and mechanical properties significantly decreased as carbon fiber and graphite powder decreased and increased, respectively. Based on these results, when manufacturing a thermoplastic resin composite reinforced with carbon fibers and graphite powder with different shapes simultaneously, properties may vary depending on the length of the carbon fiber and the diameter of the graphite powder. However, the hybrid system has a golden ratio with homogeneous filler dispersion.

## Figures and Tables

**Figure 1 polymers-14-01935-f001:**
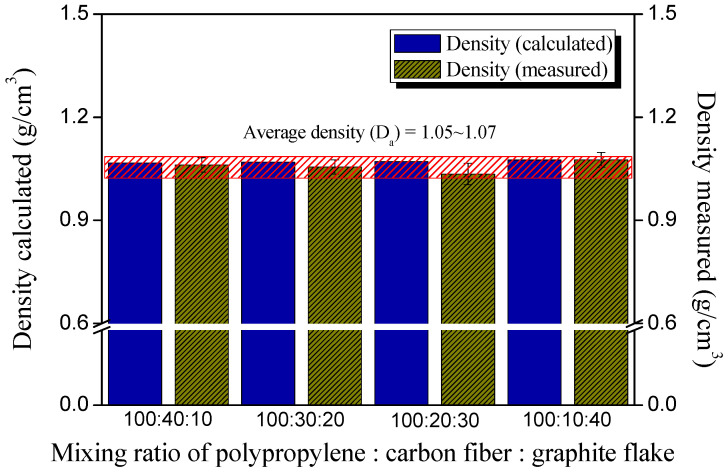
Calculated and measured densities of the composites with different mixing ratios of polypropylene:carbon fibers:graphite flake.

**Figure 2 polymers-14-01935-f002:**
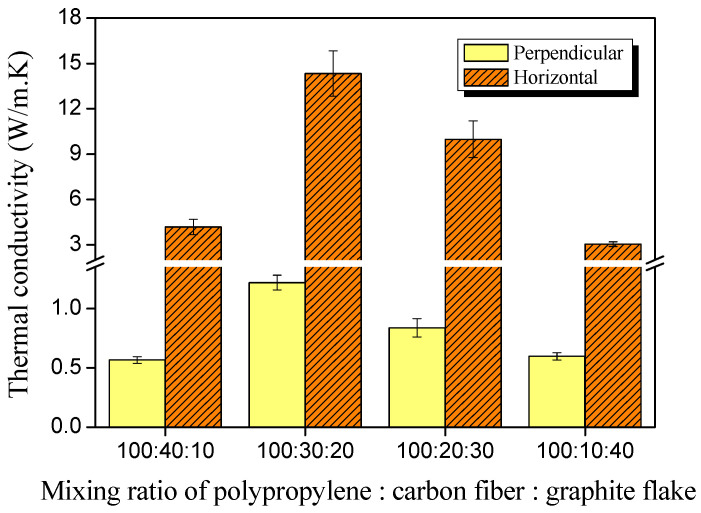
Perpendicular and horizontal thermal conductivity of the composites with different mixing ratios of polypropylene:carbon fibers:graphite flake.

**Figure 3 polymers-14-01935-f003:**
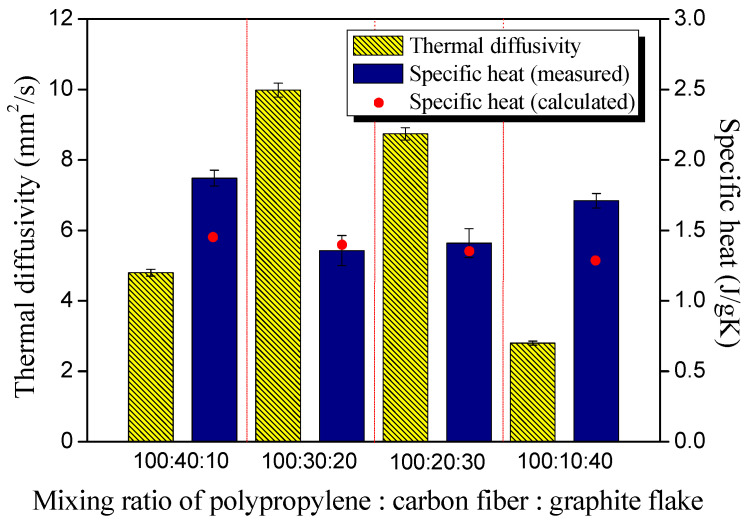
Thermal diffusivity and specific heat values of the composites with different mixing ratios of polypropylene:carbon fibers:graphite flake (Specific heat values of reference PP, CF, and GP are 1.8, 0.75, and 0.7 J/gK, respectively).

**Figure 4 polymers-14-01935-f004:**
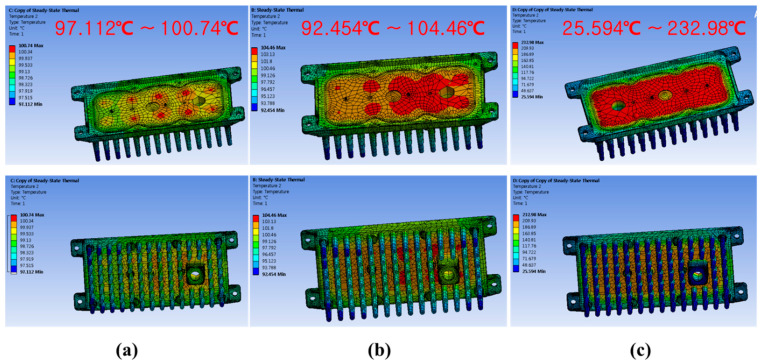
Thermal analysis simulations of product design; (**a**) aluminum materials, (**b**) polypropylene composites with hybrid carbonaceous fillers applied, (**c**) polymer materials.

**Figure 5 polymers-14-01935-f005:**
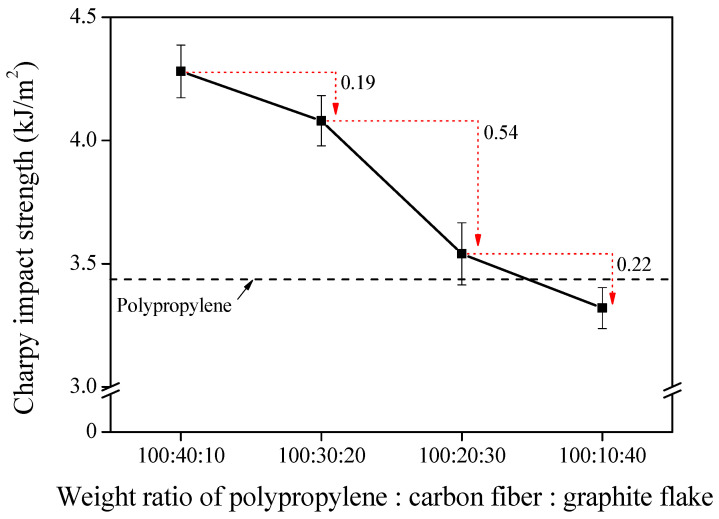
Charpy impact strength of the composites with different mixing ratios of polypropylene:carbon fibers:graphite flake.

**Figure 6 polymers-14-01935-f006:**
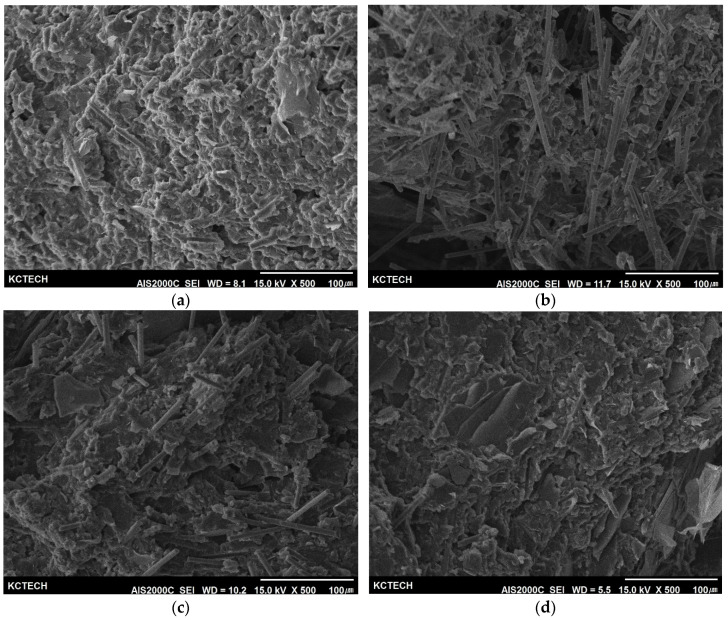
Cross-section SEM images of the composites with different mixing ratios of polypropylene:carbon fibers:graphite flake; (**a**) 100:40:10, (**b**) 100:30:20, (**c**) 100:20:30, (**d**) 100:10:40.

**Table 1 polymers-14-01935-t001:** Typical thermophysical properties of the materials used.

Samples	Average Size	Density * (g/cm^3^) @ 25 °C	Specific Heat * (J/g·K)	Thermal Conductivity * (W/mK)
Polypropylene	-	0.9	1.8	0.12~0.2
Carbon fibers	L: 1 inch, D: 7 μm	1.8	0.75	5~10
Graphite flakes	W: 150 μm, T: 25 μm	1.9	0.7	200~250

(L: length, D: diameter, W: width, T: thickness). * These values are provided from the technical sheet of the supplier.

**Table 2 polymers-14-01935-t002:** PP/CF/GF formulation for the composite preparation.

Sample Name	PP (g)	CF (g)	GF (g)	W_f_ * (%)	V_f_ ** (%)
100:40:10	100	40	10	33.3	19.83
100:30:20	100	30	20		19.66
100:20:30	100	20	30		19.49
100:10:40	100	10	40		19.32

* W_f_: Weight fraction of fillers, ** V_f_: Volume fraction of fillers.

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
