# Peer review of "Effects of Mixing Ratio of Hybrid Carbonaceous Fillers on Thermal Conductivity and Mechanical Properties of Polypropylene Matrix Composites"

_polymers, 2022, doi:10.3390/polym14101935_

Round 1
Reviewer 1 Report
The comments of polymers-1684170 were listed as follows:
(1)The properties of pure PP is missing. Please provide theese data accordingly. This will help the readings to understand the role of inorganic fillers played in the composite systems.
(2) Although it might be beyond the scope of this work, how is the other mechanical properties such as tensile and flexural strength and modulus? In other words, will carbon fiber and graphite flake have a reinforcing effect or not?
(3) There might be many small pieces in the section of Introduction. I suggested the authors revise the Introduction part thoroughly.
Thanks.
Author Response
Thank you for your kind comments.
I have attached the answer to the comments.

Reviewer 2 Report
It is an interesting and practical research.
I recommend it for publication.
Author Response
Thank you for your kind comments.

Round 2
Reviewer 1 Report
The authors have addressed my comments.